# Insights from Avian Influenza: A Review of Its Multifaceted Nature and Future Pandemic Preparedness

**DOI:** 10.3390/v16030458

**Published:** 2024-03-17

**Authors:** Jianning He, Yiu-Wing Kam

**Affiliations:** Division of Natural and Applied Science, Duke Kunshan University, No. 8 Duke Avenue, Kunshan 215316, China; jianning.he@dukekunshan.edu.cn

**Keywords:** avian influenza, pandemic management, epidemiology, virology, pathogenesis, antiviral drugs, influenza vaccines, immune response, antigenic drift, antigenic shift

## Abstract

Avian influenza viruses (AIVs) have posed a significant pandemic threat since their discovery. This review mainly focuses on the epidemiology, virology, pathogenesis, and treatments of avian influenza viruses. We delve into the global spread, past pandemics, clinical symptoms, severity, and immune response related to AIVs. The review also discusses various control measures, including antiviral drugs, vaccines, and potential future directions in influenza treatment and prevention. Lastly, by summarizing the insights from previous pandemic control, this review aims to direct effective strategies for managing future influenza pandemics.

## 1. Introduction

Influenza is a common illness that affects a great number of people around the globe. Historical accounts of influenza pandemics date back to ancient times, with Hippocrates and Livy mentioning them as early as 412 B.C. [1]. Starting in the 17th century, as the disease gained widespread recognition and investigation, influenza received significant attention from both scientific and lay publications [2]. During the Russian influenza pandemic of 1889, Haemophilus influenzae was mistakenly identified as the cause of influenza, but this paved the way for modern medicine to be used to study influenza [3]. Following that, a British physiologist named Walter Morley Fletcher isolated the influenza virus as the causative agent of flu in 1933 [3]. Until the 21st century, seasonal influenza still affected about 10% of the global population, and nearly half a million people still die from the disease each year globally [4].

The causative agent of influenza is the influenza virus. This family contains four genera: Influenza A virus, influenza B virus, influenza C virus, and influenza D virus [5]. Both influenza A and B viruses have a lipid envelope containing two surface proteins: hemagglutinin (HA) and neuraminidase (NA) [6], while influenza C and D viruses have hemagglutinin–esterase–fusion (HEF) glycoproteins on their surfaces [7]. In addition, both the influenza A and B virus genomes contain eight negative-sense, single-stranded RNA segments, while the influenza C and D virus genomes contain seven segments instead [7].

Among the four genera of influenza viruses, influenza A virus is regarded as the most important type of influenza virus to be studied due to its potential to cause severe pandemics with high morbidity and mortality rates [8]. Humans have experienced several severe pandemics caused by the influenza A virus, including the Spanish Flu in 1918, the Asian flu in 1957, the American swine flu in 1976, and the 2009 influenza pandemic caused by the swine-origin reassortant virus (pH1N1) [3,8,9]. Influenza A viruses also have high mutation rates, resulting in their unpredictability and the ability to cause new epidemics or pandemics [8]. Some of the influenza A virus genus is known to infect birds and is called avian influenza. This review will mainly focus on the characteristics of the influenza A virus and its AIV subtypes.

The RNA genome segments in the influenza A virus encode eight different viral proteins. These include three membrane-bound proteins responsible for antigen–antibody interaction: hemagglutinin, neuraminidase, and matrix protein 2 (M2) [10]. Glycoprotein HA facilitates viral entry into host cells by binding to sialic acid receptors on the surface of target cells, whereas NA is essential for viral particle release from infected cells [8]. Ion channel M2 acidifies the interior of the viral particle, causing the viral ribonucleoprotein complex to dissociate from the matrix protein and allowing the viral genome to enter the host cell [8]. In the ribonucleoprotein core, polymerase basic protein 1 (PB1), polymerase basic protein 2 (PB2), polymerase acidic protein (PA), and NP form polymerase complexes responsible for viral replication [10]. Additionally, nuclear export protein/non-structural protein 2 (NEP/NS2) and matrix protein 1 (M1) exist in the virus, while non-structural protein 1 (NS1) is not packaged in virions [10]. The function of NEP/NS2 and NS1 is to promote viral RNA replication, whereas M1 has multiple functions, such as providing structural support and stability to the virus particle, facilitating viral assembly, participating in viral RNA synthesis regulation, etc. [8].

The influenza A virus subtypes can be differentiated by the H and N antigens on the virus surface [11]. Each influenza virus expresses one H antigen and one N antigen, which can appear in any combination [11]. Currently, 18 different H subtypes (H1–H18) and 11 N subtypes (N1–N11) have been identified [12]. The first report of an avian influenza infection was in northern Italy in 1878 [13]. Researchers discovered that the pathogen of avian influenza was an influenza A virus subtype by identifying the presence of type A influenza virus type-specific ribonucleoprotein, and it was officially named avian influenza virus in 1955 [13]. Different strains of avian influenza have varying levels of lethality and infectivity for humans and birds. Most of the avian influenza viruses belong to low-pathogenic avian influenza (LPAI) [11]. Only specific subtypes carrying H5 or H7 with a polybasic HA cleavage site belong to highly pathogenic avian influenza (HPAI), which is highly dangerous to susceptible species, including birds and humans [11,14]. Apart from HPAI, some H9 subtypes of LPAI can also cause human infection and have pandemic potential, such as the cases of H9N2 in China (Figure 1).

The first reported HPAI was in Scotland in 1959 [13]. Over the years, HPAI outbreaks have occurred in various countries and regions, including North America, Europe, the Middle East, and East and Southeast Asia (Figure 1). AIV outbreaks have occurred globally, with the highest frequencies in Asia, especially China (Figure 1). One of the prevalent avian influenza strains in Asia is human avian influenza H7N9. Human infections of H7N9 were first reported in China in 2013 and have continued to occur almost annually, with a total of 1568 reported infections and a high mortality rate (Figure 1). The occurrence of AIV outbreaks persists throughout the year, with a peak in frequency from November to May [15].

**Figure 1 viruses-16-00458-f001:**
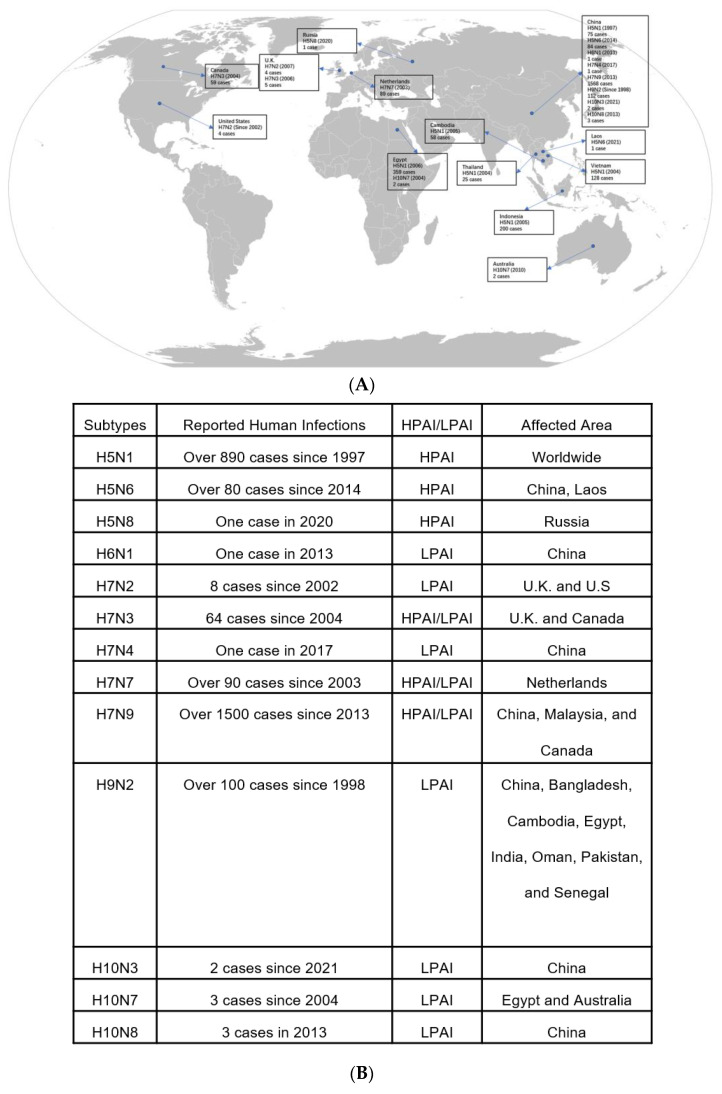
Global distribution of AIV infections among humans from 1997 to 2023 [14,16,17,18,19,20,21,22,23,24,25]. (**A**) The world map illustrates the geographical distribution of human cases of various subtypes of avian influenza viruses reported from 1997 to 2021. Each marker on the map corresponds to a country, with the number of cases and the year of the first reported case beside it. (**B**) A summary table categorizes the avian influenza virus subtypes based on reported human infections, the pathogenicity of the virus (classified as either HPAI or LPAI), and the affected areas. The table provides a quick reference to the spread and severity of each subtype, showing the total number of human cases reported since the year the first case was detected for each subtype, as well as the geographical regions where these cases were reported.

## 2. Virus Dissemination

Because it involves the transmission of potentially zoonotic and pandemic viruses between different hosts, such as wild birds, poultry, and humans, the spread of AIVs is a major topic in public health and animal health. Avian species, particularly waterfowl and shorebirds, serve as natural reservoirs for AIVs to grow and mutate [26].

### 2.1. Global Spread of AIVs

The dissemination of AIVs within and across regions is largely influenced by the migratory movements of wild birds, which can carry and exchange different virus strains along their flyways [27]. Flyways are large corridors that connect migratory birds’ breeding and wintering grounds across continents [27]. Studies have shown, for example, that the Pacific Flyway, a migratory path connecting western North America to eastern Asia via breeding grounds in Beringia, has the potential to introduce or exchange avian influenza viruses between North America and Asia [27].

### 2.2. Regional and Local Spread of AIVs

AIVs can spread and exchange between host species and populations through various pathways, including direct or indirect contact with infected live birds or corpses, inhalation of virus-containing aerosols and respiratory droplets, and sharing of contaminated equipment [16,28].

The live poultry market plays a crucial role in the spread of AIVs, as it is a place where various bird species and populations come into contact. These markets could be a source of virus transmission between domestic and wild bird populations, as well as between humans and birds [16]. Apart from live poultry markets, commercial poultry farms, family backyard farms, and zoos are all places for AIVs to reproduce, mutate, and spread among birds, mammals, and humans [29].

### 2.3. Antigenic Drift

Antigenic drift is the main cause of the AIV mutation [26]. Antigenic drift is a gradual evolutionary process driven by the selective pressure to evade host immunity that occurs as a result of frequent mutations within the antibody-binding sites of the influenza virus’s HA and NA proteins [30]. The mutations reduce host antibody binding, allowing the virus to spread more efficiently and widely among the population [30]. For example, the H3N2 subtype of the influenza A virus has exhibited antigenic drift since it was introduced into humans in 1968 [31]. These mutations can lead to altered antigenic properties of hemagglutinin and neuraminidase, causing the virus to be recognized differently by the immune system, thus escaping neutralization by pre-existing antibodies [31]. Some antigenic variants resulting from drift may possess increased fitness or enhanced pathogenicity, leading to more severe illness or higher rates of transmission [31]. This necessitates the regular update of influenza vaccines to match the predominant circulating strains and maintain their effectiveness [31].

### 2.4. Antigenic Shift

Antigenic shift is another cause of mutation due to the segmented structure of the genome of influenza A viruses [32]. It happens when several different influenza A viruses, each containing a different combination of gene segments, infect the same host cell [32]. The gene segments of the different viruses can mix and match during replication, producing new viral particles with novel combinations of genes, including changes in their HA and NA proteins [32,33]. The new virus, after an antigenic shift, may have the ability to infect a new host, such as a human [32]. Since the human immune system does not recognize the novel virus, it can spread rapidly through the population, increasing the risk of a pandemic [32]. Several pandemics in history have been proven to be caused by antigenic shifts, such as the Spanish flu in 1918, the H5N1 outbreak in Hong Kong in 1997, and the H1N1 pandemic that happened worldwide in 2009 [9,32,33]. The H1N1 influenza A virus responsible for the Spanish flu and 2009 pandemic had a novel combination of HA and NA genes, which likely emerged through an antigenic shift involving reassortment between swine, human, and avian influenza viruses over several years [9,34].

### 2.5. The Mechanism of Human Infections

Influenza virus enters the host cell by first recognizing a terminal α-sialic acid that is linked to saccharides anchored on the host cell surface, which serves as a specific receptor molecule, allowing it to gain access to the host cell where it can replicate [17]. The virus particle then induces endocytosis, which creates an endosome that encapsulates the virus [35]. The endosome is transported near the nucleus, and the virus membrane fuses with the host membrane, releasing the eight segments of the RNA genome into the nucleus [17]. This initiates virus transcription and replication, which allow the virus to propagate within the host cell [17]. After binding to and entering the targeted epithelial cells, the virus can cause damage to them, resulting in cell death and viral particle shedding into the airway, resulting in fever, cough, and sore throat symptoms [35]. The infection mechanism will trigger different levels of pathogenic effects in human bodies, which will be discussed in the next section.

## 3. Clinical Features and Severity of Disease

AIVs primarily affect birds, while a small number of strains of over 100 subtypes currently discovered can infect humans and cause respiratory illness with varying clinical features and severity [36]. However, since 2013, a newly emerged strain of AIV, known as H7N9, has caused over 1500 human infections and 600 deaths in China, while another strain, H5N1, has resulted in over 800 human cases and 400 deaths worldwide since its initial emergence in Hong Kong, China [21,36].

The severity of symptoms can vary depending on the human’s immune response and other factors. Human infection with AIVs ranges from mild to severe, with mild cases of influenza-like illness being more common [37]. Signs and symptoms of a mild AIV infection include nasal obstruction, cough, sore throat, fever, shortness of breath, etc. [37]. In these mild cases, individuals often experience a self-limiting illness and recover within a week [37].

For severe human infections, usually caused by HPAI strains like H7N9 or H5N1, it typically takes 5–7 days from initial onset to the development of severe conditions such as ARDS and severe pneumonia [38,39]. Severe cases have shown symptoms like significant difficulty breathing and low oxygen levels [38,39]. Patients showed signs of recovery after receiving intensive care for about 23–24 days [39]. The outbreaks of H7N9 and H5N1 in China have an especially high case-fatality rate [39]. The data show that the rate is 18.7% for H7N9 and 59% for H5N1 [39]. In addition, the severity of AIV infection can be different for some other factors, including age, immune status, and viral load [40]. Patients with pre-existing health conditions such as diabetes, cardiovascular disease, or respiratory diseases are at higher risk of developing severe AIV infection [40]. H5N1 infection may also result in viremia at high levels and for prolonged periods [41]. This allows the virus to spread to extra-respiratory tissues, such as the brain, intestine, liver, lymph nodes, spleen, bone marrow, placenta, fetus, and kidneys [41]. These tissues often exhibit associated lesions, including edema, demyelination, necrosis, accumulation of reactive histiocytes, hepatic necrosis, lipidosis, cholestasis, and hemophagocytic activity in lymph nodes [41].

Humans infected with avian influenza may exhibit rare or atypical symptoms in addition to the typical respiratory signs, including gastrointestinal upset (diarrhea and vomiting) and neurological symptoms (headache, confusion, seizures, and altered mental status) [38,41,42]. Cardiovascular manifestations, such as myocarditis, have been observed in some severe cases. Additionally, coagulation abnormalities and disseminated intravascular coagulation (DIC) have been reported, which can result in abnormal bleeding and organ damage [42,43].

## 4. Characteristics of AIVs

Avian influenza viruses, belonging to the influenza A virus family, are characterized by several unique features in their genetic makeup and protein composition, which play crucial roles in their replication and interaction with host cells [10]. AIVs is composed of eight segments of single-stranded negative-sense RNA, encoding for different viral proteins [10].

Among the viral proteins, hemagglutinin (HA) is significant in the life cycle of avian influenza viruses (Figure 2). HA is responsible for mediating the entry of the virus into host cells and determining the host range and tissue tropism of the virus [44]. The HA protein binds to specific receptors on the surface of host cells, primarily sialic acid receptors, allowing the virus to enter the correct cell types [44]. Avian influenza viruses typically show a preference for α(2,3)-linked sialic acid receptors (NeuAcα2-3Gal), which are present in bird trachea and intestines [44,45]. This receptor specificity restricts the natural transmission of avian influenza viruses to certain species. However, the genetic changes in the HA protein can enable the virus to switch its receptor binding preference to α(2,6)-linked sialic acid receptors (NeuAcα2-6Gal), which are predominantly found in the human upper respiratory tract [44]. Such receptor-binding adaptations are critical for avian influenza viruses to cross the species barrier and potentially infect humans and other mammals.

In addition to HA, another important protein in the life cycle of avian influenza viruses is neuraminidase (NA) (Figure 2). NA acts as an enzyme that cleaves sialic acid residues, which are present on the surface of host cells and are also used by HA to attach to [44]. During the later stages of viral replication, newly formed influenza viruses bud from the host cell membrane. However, the viral particles remain attached to the infected cell surface through their binding to sialic acid receptors [44]. This attachment can hinder the release of viral particles and limit the spread of the virus to neighboring cells and tissues [44]. NA can facilitate the release of newly formed viral particles from infected cells and prevent viral aggregation by cleaving sialic acid residues from glycoproteins on the host cell membrane, thus helping viruses to infect more healthy cells [44].

The RNA-dependent RNA polymerase and NP of AIVs can contribute to the pathogenicity of HPAI in host bodies [46]. The HPAI virus polymerase’s adaptation to interact with proteins in mammalian hosts is thought to enhance the synthesis of viral RNA, which can increase the severity of the disease [46]. For example, the E627K, D701N, and Q591K mutations in the PB2 protein have been associated with increased virulence and the adaptation of HPAI to mammalian hosts [46].

## 5. Role of Immune System in the Pathology of AIVs

### 5.1. Innate Immunity

The innate immune response is the first line of defense against AIV infection. The initial response of the innate immune system to the avian influenza virus triggers an increase in the production of cytokines and chemokines, especially within respiratory epithelial and primary endothelial cells, resulting in a significant rise in proinflammatory cytokines like TNF-α and IFN-β [47]. Notably, infections with H5N1 can induce a “cytokine storm,” characterized by unusually high levels of proinflammatory cytokines and chemokines, contributing to disease pathogenesis [47]. Despite a strong induction of type I interferons and other acute phase response genes, this intense immune response often fails to control the rapidly progressing infection, as observed in studies with human cells and animal models [47].

### 5.2. Adaptive Immunity

Adaptive immunity develops late and is specific to the influenza virus strain. The adaptive immune response is composed of two main branches: humoral immunity, which involves virus-specific antibodies, and cellular immunity, which includes virus-specific CD4+ and CD8+ T cells [48].

#### 5.2.1. Humoral Immunity

There are two kinds of humoral immune responses: primary and secondary B-cell responses. The primary response involves both innate-like and conventional B-1 cells, producing IgM antibodies [49]. During the infection’s progression, B-2 cells produce IgG, IgA, and IgE antibodies tailored to specific influenza antigens and have higher affinity compared to IgM antibodies [49]. During secondary B cell activation, which occurs upon subsequent exposure to the same or similar influenza strains, memory B cells are rapidly activated to produce high-affinity antibodies more quickly and in greater quantities [50].

The humoral immune response against avian influenza viruses is characterized by antibody (Ab) responses, particularly against the hemagglutinin (HA) protein of the viruses [47]. The initial infection triggers the production of IgM, IgA, and IgG antibodies [48]. The secondary exposures predominantly involve IgA and IgG responses, with IgM typically absent [48]. A key aspect of antibody response, especially in mucosal areas like the respiratory tract, is the production of secretory IgA (sIgA) antibodies. sIgA is produced by plasma cells in mucosal tissues and secreted into the lumen [51]. It provides frontline defense by neutralizing pathogens at mucosal surfaces, preventing their entry into the body [51]. It effectively clears viruses from infected epithelial cells and can redirect antigens from the lamina propria to the lumen without triggering inflammatory responses [51].

The humoral immune response produces antigens that target specific epitopes, including HA, NA, M2, and NP viral proteins [48]. It is important to find out the conserved epitopes of the influenza virus. By focusing on conserved epitopes, such as those found in the hemagglutinin stalk and the M2 protein, vaccines can potentially provide protection against a wide range of influenza strains, reducing the need for frequent vaccine updates and offering better defense against pandemic and seasonal influenza outbreaks [52].

#### 5.2.2. Cell-Mediated Immunity

T cells are critical for controlling and clearing AIV infections. CD8+ cells recognize highly conserved viral peptides, primarily from internal and conserved viral proteins, allowing for broad cross-reactive immunity even against novel avian influenza viruses [47]. Individuals recovering from avian influenza infections, such as H7N9, show robust IFN-γ+CD8+ T cell responses, contrasting with individuals who succumb to the infection, who exhibit fewer of these responses and more prolonged activation of exhausted T cells [47]. Recovery is often marked by the robust expansion of cross-reactive CD8+ T cell clonotypes, highlighting the critical role of influenza-specific CD8+ T cells in mediating recovery [47]. Additionally, CD4+ T cells and mucosal-associated invariant T (MAIT) cells also contribute to the response, with cross-reactivity toward avian viruses and potential roles in promoting recovery through the production of antiviral cytokines [47].

Naive CD8+ T cells can differentiate into CTLs and destroy infected cells, providing immunity against different strains of influenza, as shown in historical patterns of infection and immunity [48]. For example, people who experienced symptomatic H1N1 infection before the 1957 pandemic showed some resistance to the H2N2 strain [48]. The reason for this cross-reactivity against various strains is because of the targets of CTLs, which are highly conserved M1, NP, PA, and PB2 viral proteins [48].

Regulatory T cells (Tregs) and T helper 17 cells (Th17) also play an important role in cell-mediated immunity against influenza viruses [48]. Tregs can regulate immune responses to prevent excessive tissue damage, while Th17 is essential in countering secondary bacterial infections, such as *S. aureus* pneumonia [48].

## 6. Control Measures

Management and control of AIVs potentially involve several measures, from early detection of cases and case alerts and quarantine measurement to antiviral drugs and vaccine development. Apart from the measurements for humans, large-scale culling, sanitizing poultry markets, and applying avian vaccination were successful in controlling AIV outbreaks.

### 6.1. Antiviral Drugs

For treating avian influenza, the antiviral drugs oseltamivir (Tamiflu), zanamivir (Relenza), and peramivir are frequently used [53]. These neuraminidase inhibitors, along with Polymerase Inhibitors (baloxavir), are generally effective against most strains of AIVs, including H7N9, H5N1, and H5N6 [53]. However, these strains often show resistance to adamantanes, which include amantadine and rimantadine [53].

#### 6.1.1. Neuraminidase Inhibitors

Neuraminidase inhibitors, such as oseltamivir (Tamiflu) and zanamivir (Relenza), are commonly used antiviral drugs for the treatment of influenza [54]. They work by inhibiting the NA on the surface of the influenza virus (Figure 2) [54]. By preventing the cleavage of sialic acid receptors, these drugs can cause the accumulation of viral particles on the infected cell surface [54]. As a result, the spread of the virus is impeded, reducing the severity and duration of influenza symptoms.

#### 6.1.2. M2 Ion Channel Inhibitors

The M2 protein forms an ion channel in the viral envelope, facilitating the viral genome to enter the host cell [8]. M2 ion channel inhibitors, including amantadine and rimantadine, target the M2 protein of influenza A viruses (Figure 2) [55]. They block the function of the M2 protein by binding to the pore of the ion channel, which prevents the influx of protons into the viral particle, disrupting the uncoating process and subsequent viral replication [55]. This inhibition of the M2 ion channel effectively reduces the production of infectious viral particles. However, the widespread emergence of resistant strains has significantly limited the usefulness of M2 ion channel inhibitors in recent years [56].

#### 6.1.3. Polymerase Inhibitors

Polymerase inhibitors are a newer class of antiviral drugs that target the viral polymerase complex involved in viral replication. The main polymerase inhibitors currently approved for influenza A virus treatment are baloxavir marboxil, pimodivir, and favipiravir, while favipiravir and baloxavir marboxil are also effective on influenza B viruses [57]. Baloxavir marboxil targets the cap-dependent endonuclease activity of the polymerase acidic (PA) protein of influenza A and B viruses (Figure 2) [57]. The PA protein plays a crucial role in viral replication by cleaving host mRNA molecules in order to “cap-snatch” the capped fragments and produce primers for viral mRNA synthesis [58]. Once inside the host cell, baloxavir marboxil is converted to its active form, which is baloxavir acid [58]. Baloxavir acid binds to the conserved active site of the PA protein and inhibits its endonuclease activity [58]. By blocking the endonuclease activity, baloxavir prevents the cleavage of host mRNA, disrupting the cap-snatching process and inhibiting viral mRNA synthesis [58]. Pimodivir is an inhibitor of the polymerase basic protein 2 (PB2) of influenza A viruses (Figure 2) [59]. It works by preventing the binding of PB2 to the 7-methyl GTP cap structures of host mRNA, thereby inhibiting the early stages of viral transcription [59]. Favipiravir targets viral RNA-dependent RNA polymerase (RdRp) and introduces errors in the genetic code of the virus (Figure 2) [60]. These mutations can lead to the generation of non-functional or less viable viral particles, thus reducing the viral load in the body [60]. Favipiravir also has broad-spectrum antiviral activity against RNA viruses like influenza, Ebola, Rabies, etc. [60]. Its precise antiviral mechanism and effects are still being investigated [60].

Apart from the antiviral drugs mentioned above, there are some new potential development directions for influenza antiviral therapy. First, utilizing anti-influenza virus antibodies as a therapeutic approach shows promise in preventing the binding of virions to target cells [61]. Convalescent blood products, immunoglobulins, and monoclonal antibodies have demonstrated efficacy in protecting animals from lethal infections and improving the condition of severely ill patients [61]. Second, sequence-based therapies involving antisense DNA oligomers and short interfering RNA (siRNA) molecules targeting viral mRNA hold the potential to block viral replication and protein synthesis [61]. What is more, combining drugs with different mechanisms of action could enhance antiviral effects and reduce the risk of drug resistance [61]. Combinations of adamantanes and NA inhibitors, like rimantadine plus zanamivir, have shown synergistic effects in animal models [61]. Therefore, exploring these alternative approaches in influenza antiviral therapy holds great promise for the future.

### 6.2. Vaccines

There are mainly three ways of vaccine production: egg-based, cell culture-based, and recombinant [62,63]. Egg-based influenza vaccines are commonly produced using fertilized chicken eggs as a substrate for growing the influenza virus. These vaccines can be either inactivated (containing dead viruses post-chemical treatment like formaldehyde or beta-propiolactone) or attenuated (containing weakened live viruses) [62,64]. Egg-based vaccines have a potential risk for individuals with severe egg allergies. However, advances in vaccine manufacturing have significantly reduced the amount of egg protein present in the final vaccine product, making it generally safe for most people with egg allergies [62]. Cell culture-based vaccines, a more recent development, are produced by growing the influenza virus in mammalian cell cultures [63]. This method avoids the egg-adapted mutations that can occur in egg-based vaccines, potentially providing a closer antigenic match to circulating strains of the influenza virus [63]. However, challenges such as the requirement of high-yielding re-assorted viruses and potential genetic changes during production may affect the vaccine’s effectiveness [63,65,66]. Recombinant influenza vaccines (RIVs) are produced using recombinant DNA technology, where specific genes encoding viral proteins are inserted into host cells, such as insect cells or mammalian cells [62,65]. The host cells then produce large quantities of the desired viral proteins to make the vaccines [62]. Although recombinant protein vaccines are safer, faster, and cheaper than egg-based vaccines, they also face challenges similar to those of cell culture-based vaccines [65,66].

#### 6.2.1. Live Attenuated Influenza Vaccines

Live attenuated influenza vaccines (LAIV) are derived from live influenza viruses that have been modified to be less virulent [62]. LAIV can simulate the natural infection process, leading to the production of both IgA and IgG antibodies without leading to severe adverse reactions [67]. LAIVs for humans against avian influenza are available in certain regions, including the US, Canada, and Europe [67].

#### 6.2.2. Inactivated Influenza Vaccines

Inactivated influenza vaccines are formulated using inactivated influenza viruses that are grown in eggs or cell cultures [68,69]. Egg-based, inactivated veterinary influenza vaccines are the most commonly used type of avian influenza vaccine in China’s poultry industry [67].

#### 6.2.3. Subunit Influenza Vaccines

Subunit influenza vaccines contain specific components of the influenza virus. These vaccines offer a high level of safety as they do not contain infectious viruses [70]. Promising targets for subunit vaccines against influenza viruses include viral structural proteins like M2e, HA, and NP [69]. In addition, proteins like M1 and NA hold promise for future development as subunit vaccines against influenza [69]. Flublok, a kind of recombinant protein subunit vaccine, has been approved by the FDA and used for many years [62]. Flublok is an influenza vaccine that contains full-length HA protein antigens derived from three selected influenza virus strains [65]. These antigens are produced in a “non-transformed, non-tumorigenic continuous cell line from Sf9 cells of the fall armyworm”, and they are “full-length proteins containing the transmembrane domain and the HA1 and HA2 regions” [65]. The vaccine aims to induce the production of HA inhibition (HAI) antibodies and prevent influenza infection [65].

#### 6.2.4. Epitope-Based Influenza Vaccines

Epitope-based influenza vaccines target specific antigenic sites on the influenza virus that are conserved and immunogenic regions, reducing the impact of antigenic variation and potentially providing broader and longer-lasting protection against diverse influenza strains [71]. Epitope-based influenza vaccines can stimulate the immune system to produce antibodies against critical epitopes on the viral proteins [72]. Potential targets include conserved regions of the HA, NP, M1, and M2 proteins of influenza viruses [71,72]. Several epitope-based influenza vaccine candidates are in various stages of development, such as “M001 (BiondVax Pharmaceuticals Ltd., Jerusalem, Israel), Flu-V (hVivo, London, UK), FP-01.1 (Immune Targeting Systems Ltd., London, UK), and rMVA-k1-k2 (Federal Medical–Biological Agency, Moscow, Russia)” [72].

#### 6.2.5. mRNA Influenza Vaccines

The mRNA influenza vaccine is a new approach to influenza immunization, leveraging the same technology as the highly successful COVID-19 mRNA vaccines [73]. It works by introducing a small piece of synthetic messenger RNA (mRNA) into the body, encoding specific virus proteins, such as the HA protein of influenza viruses [73,74]. Once inside cells, the mRNA instructs them to produce harmless fragments of the target protein, inducing an immune response and antibody production [74]. mRNA technology, proven successful in COVID-19 vaccines, offers hope for more effective influenza vaccines [73]. It eliminates egg-based production, potentially improving strain matching and manufacturing [73]. Clinical trials have shown promise, although robust seroconversion across strains remains a challenge. mRNA allows multiple antigen combinations, potentially broadening protection [73]. Recent research expanded this to a quadrivalent influenza A group 2 mRNA formulation, offering protection for multiple influenza A viruses, which demonstrates the broad protection of mRNA vaccines [73]. Currently, mRNA influenza vaccines from Moderna and Pfizer have begun phase 3 clinical trials, and vaccines from Sanofi/Translate Bio and GlaxoSmithKline (GSK)/CureVac are under phase 1 [73].

## 7. Lessons Learned from the Avian Influenza and Strategies for Future Pandemic

This review summarizes the multifaceted nature of avian influenza, covering its epidemiology, virology, hosts, vectors, symptoms, immune responses, and treatments. The ability of the virus to mutate, including antigenic drift and shift, enhances its ability to evade host immunity and adapt to new hosts. Vaccines and antiviral drugs are the two main ways to combat AIV infection. However, the influenza virus can co-evolve with preventive and treatment measures, making the control of the disease difficult.

Looking back on the COVID-19 pandemic, we have learned a lot about controlling this airborne and zoonotic virus that can be applied to future pandemics. Future approaches emphasize developing new models for public sector funding and research, increasing testing capacity with new technologies, and building a connected diagnostic system for surveillance [75]. Addressing social inequalities is highlighted as a critical intervention. Control measures like social distancing can have a negative impact on the most vulnerable groups in society, widening socio-economic gaps between the richest and poorest. Future policies should consider the interests of vulnerable groups to increase the overall effectiveness of policies and minimize their negative impacts [76]. Additionally, understanding the current challenges and potential limitations of COVID-19 vaccination in this population is critical, not only to inform decisions about additional vaccine doses but also to advise on non-pharmacologic prevention strategies and prophylactic approaches. Because of the potential for COVID-19 to transition from a pandemic to an endemic, establishing effective healthcare interventions to protect immunocompromised individuals from the ongoing threat of infection remains a public health priority [77]. Adapting and applying these approaches will enable us to better prepare for and respond to the risk of future outbreaks, including potential influenza pandemics.

Here, we propose some possible AIV control measures in two aspects: biology and public policies. Firstly, developing avian-specific vaccines and antiviral drugs is crucial to preventing the spread of influenza among birds. Concurrently, advancing human vaccines, particularly leveraging rapid advancements in mRNA technology, is essential for more effective and timely protection. Secondly, it is important to establish and enhance surveillance systems for wildlife and livestock diseases, providing early warning for potential influenza risks. Implementing stringent laws to limit wild bird hunting and live bird market activities can reduce human–bird interaction, thereby controlling infection sources. Finally, intensifying public awareness and education, particularly about the risks of consuming poultry during high-risk influenza periods, is vital for preventing infections.

In conclusion, comprehensive strategies are needed to prevent and respond to future pandemics caused by AIVs. The joint efforts of governments across the world and of various health organizations will enable the effective implementation of these strategies and achieve even better results.

## Figures and Tables

**Figure 2 viruses-16-00458-f002:**
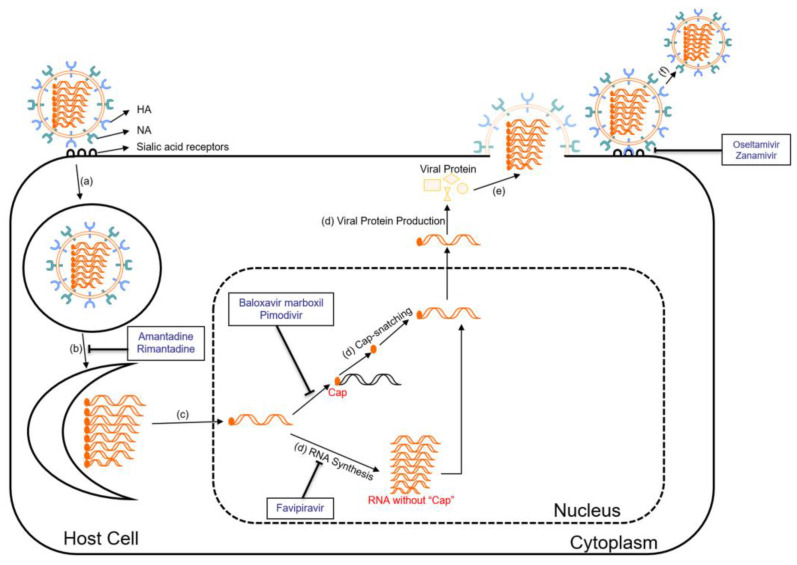
Life cycle of influenza virus and sites of action of antiviral drugs. The influenza virus life cycle involves a series of steps, including (a) HA binding to sialic acid on the cell surface, leading to endocytosis; (b) M2 protein facilitating membrane fusion using M2 ion channels; (c) the viral RNA and RdRp complex releasing into the cytoplasm through fusion, followed by transportation to the nucleus for replication; (d) viral mRNA synthesis and Cap-snatching, forming viral mRNA, leading to viral protein production; (e) viral proteins and genomic RNA move to the cell surface assemble into the new virion, bud from the membrane, and form new viral particles; (f) neuraminidase (NA) cleaves the sialic acid-HA bond, releasing virus particles from infected cells for the next round of infection (the functions and mechanisms of antiviral drugs against influenza viruses will be discussed in Section 6).

## Data Availability

No new data were created in this review article.

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
