# Peer review of "Insights from Avian Influenza: A Review of Its Multifaceted Nature and Future Pandemic Preparedness"

_viruses, 2024, doi:10.3390/v16030458_

Round 1

Reviewer 1 Report

Comments and Suggestions for Authors

The authors review the epidemiology, virology, and pathogenesis, and treatments of avian influenza viruses. The review is well written and fairly broad. A few minor points below. 

line 27: "Until 21st century," - this reads oddly and needs reworking. Adding the word "the" would help. 

line 33:  ".. and neuraminidase (NA) [6].While ..." - a comma would be more appropriate than a full stop. 

line 37: "Among four genera influenza viruses"  - it would read better as "Among THE four .."

lines 45-88: It would be useful for the reader to describe the genetic basis for HPAI viruses and the change of polymerase specificity. 

line 126-129: The definition of HPAI is repeated from line 72-74

line 222-227: This is a bit confusing. NeuAcα2-6Gal is predominantly expressed in human upper respiratory cells (  tracheal epithelial cells). NeuAcα2-3Gal, along with NeuAcα2-6Gal, is expressed in lungs. Thus, under heavy viral load, unmutated AIV  (with a preference for NeuAcα2-3Gal) can also infect human lungs. 

Section 6. Control Measures  - I think it would be useful to describe in more detail the quarantine and culling as control measures. Large scale culling, cleaning poultry markets and avian vaccination were successful in controlling the original H5N1 outbreak in 1997 as well as the more recent H7N9 outbreak. 

Comments on the Quality of English Language

The manuscript reads very well except in the few places I have specified in the comments.   

Reviewer 2 Report

Comments and Suggestions for Authors

This comprehensive review article delves into the epidemiology, virology, pathogenicity, and countermeasures for avian influenza viruses. To understand the current prevalence of avian-origin influenza viruses and implement appropriate control measures, a detailed overview of their recent achievements and knowledge is essential. However, the information provided in the manuscript is often superficial and already outdated. While the authors primarily focus on avian influenza viruses, the inclusion of discussions on human influenza viruses in some chapters can be distracting for readers.

Several references cited by the authors are outdated, leading to misinformation within the manuscript. Examples include inaccuracies in lines 61-62 and lines 223-224, among others.

Despite the title's focus on avian influenza viruses, the authors exclusively provide general information about influenza viruses and discuss knowledge related to human influenza viruses in certain chapters, neglecting specific information on avian influenza viruses. Moreover, in chapters such as "5. Role of Immune System in the Pathology of Influenza Virus," the authors delve into general immunology without offering specific insights into avian or human influenza infections.

In the vaccine chapter, the authors mention egg-based and recombinant vaccines but overlook cell culture-based vaccines, widely used today. Notably, the challenges of recombinant protein vaccines mentioned in lines 389-391 align more with the limitations of cell culture-based vaccines, according to the referenced literature. Furthermore, the authors exclusively discuss trivalent vaccines, despite the prevalent use of quadrivalent vaccines nowadays.

On a minor note, in Figure 2, some blunt arrows inaccurately indicate sites. For example, Baloxavir marboxil inhibits endonuclease activity, so the blunt arrow should target the site before cap-snatching. Additionally, neuraminidase inhibitors should be directed at the site between HA and the host cell surface, not (f).

Reviewer 3 Report

Comments and Suggestions for Authors

This review article attempts to cover lessons learned during the decades of AIV outbreaks around the world. The title is misleading as I don't feel that many lessons are presented. Furthermore, there is little to no application of those lessons to future pandemic outbreaks or new pathogens that become endemic. I would suggest changing the title. References to COVID really do not belong in this review. This review does not focus on avian influenza. It is often difficult to determine if the authors are discussing human influenza or avian influenza in humans or in chickens.   

Line 8.....Influenza should be lowercase and "has" should be "have"

Line 20.....Historical not history

Line 81 and 82....From 15th May 2023, there have been a recent confirmed outbreaks of highly 81 pathogenic avian influenza (HPAI) in the state of Paraná, which is Brazil's primary region 82 for poultry production (Figure 1) [13] ......However, Figure 1 (A or B) does not show any cases in Brazil.

Line 85 and 86......The Brazilian government has also 85 implemented effective measures to prevent the disease spreading from wild birds to commercial herds [13].....if this is a lessons learned paper then explain what they did....don't just reference it.

Line 117 to 119....add sharing of contaminated equipment.

Line 126....viruses that contain HA cleavage sequences similar to viruses that have shown to be highly pathogenic are also classified as HPAIVs. 

Line 188....missing a period.....[37,38] The outbreaks 

Lines 240 to 248.....the antivirals presented in Figure 2 should be incorporated into the text. I see you cover this later......I suggest inserting a comment letting readers know that you will cover the antivirals in depth in Section 6.

Lines 469.......There is no discussion of public policy. The review should contain some information regarding surveillance efforts in birds and people. 

Comments on the Quality of English Language

Overall, very good. I have identified areas that need some light editing. 

Reviewer 4 Report

Comments and Suggestions for Authors

1.        In the Introduction section, mention should be made of the 2009 pandemic.

2.      «Currently, 16 different H subtypes (H1-H16) and 9 N  subtypes (N1-N9) have been identified [10].» - this is an erroneous statement. Currently, 18 hemagglutinin subtypes and 11 neuraminidase subtypes have been identified in influenza A viruses.[ For example, Webster R.G., Govorkova E.A. Continuing challenges in influenza. // Ann. N. Y. Acad. Sci. – 2014. – V. 1323. – P. 115-139.].  The authors do not mention anything about the surface proteins of bat influenza A viruses.

3.      It should be noted that influenza A viruses of subtype H9, like subtypes H5 and H7, also have pandemic potential.

4.      In section 2.4, it should be noted that the antigenic shift of influenza A viruses is possible due to the segmented structure of the genome of this pathogen. Also in section 2.4 it is necessary to give an example of the 2009 pandemic, the causative agent of which received the genes of swine and avian influenza viruses, and provide the corresponding link.

5.      The authors, at least briefly, do not address the problem of vaccination among poultry and do not provide information about veterinary vaccines.  Since the review article is about avian influenza viruses, this issue is worth mentioning.

Round 2

Reviewer 2 Report

Comments and Suggestions for Authors

Major comments: Please review the manuscript thoroughly. There are still many statements to be corrected. This thorough examination is essential to ensure the accuracy and reliability of the information presented. The comments below are just examples.

Lines 74-75: Other subtypes of LPAI may also have pandemic potential. “some subtypes of LPAI” would be more appropriate.

Lines 87-90: Numerous HPAI outbreaks persist globally, even as we move into 2024. To capture the ongoing situation more accurately, the authors should consider generalizing about the frequency of these outbreaks rather than focusing on a specific recent incident.

Lines 145-147: H5N1 outbreak in 1997 is not “pandemic”.

Lines 193-199: Please consider positioning this paragraph ahead of the one that begins with “For severe cases ~~”.

In line 212, the manuscript incorrectly cites reference 44, which does not discuss the distribution of avian-type receptors in bird organs. The authors should correct this by citing the appropriate reference, for instance, "DOI: 10.4172/1747-0862.1000026," to accurately reflect the source of their information. Additionally, it should be noted that the avian-type receptor is prevalent not only in the bird intestine but also in other organs.

Lines 395-402: The statement is incorrect; currently, there are no Live Attenuated Influenza Vaccines (LAIV) for humans against avian influenza that have been approved and available. Instead, recent research findings on the development and study of LAIV for humans against avian influenza should be included in this section to provide an accurate and up-to-date overview of the progress in this area.

Reviewer 3 Report

Comments and Suggestions for Authors

Line 126. The virus is likely not spread through the consumption of contaminated meat as most people think of meat but more likely through the consumption of undercooked by-products or offal collected at slaughter. Blood and intestine where virus can be found in high concentrations throughout the course of infection. Although rare, even if meat were contaminated, achieving proper cooking temperatures will inactive the virus.  Remove the reference to meat. 

Line 169- I am not sure if bronchi are considered part of the upper respiratory system. 

Section 3. Please identify when you are referring to birds and when you are referring to people. This constant switching is something that I struggle with as I read this manuscript. There is jump from birds to people and back to birds. An attempt to be made to clearly separate each section such that it is clear to the reader when birds are being discussed and when statements pertain to people. 
